# Deep Graph-Convolutional Generative Adversarial Network for Semi-Supervised Learning on Graphs

Nan Jia , Xiaolin Tian *, Wenxing Gao and Licheng Jiao

Key Laboratory of Intelligent Perception and Image Understanding of Ministry of Education, International Research Center for Intelligent Perception and Computation, Xidian University, Xi'an 710071, China; njia@stu.xidian.edu.cn (N.J.); wxgao@stu.xidian.edu.cn (W.G.); lchjiao@mail.xidian.edu.cn (L.J.)

* Correspondence: xltian@mail.xidian.edu.cn

**Abstract:** Graph convolutional networks (GCNs) are neural network frameworks for machine learning on graphs. They can simultaneously perform end-to-end learning on the attribute information and the structure information of graph data. However, most existing GCNs inevitably encounter the limitations of non-robustness and low classification accuracy when labeled nodes are scarce. To address the two issues, the deep graph convolutional generative adversarial network (DGCGAN), a model combining GCN and deep convolutional generative adversarial networks (DCGAN), is proposed in this paper. First, the graph data is mapped to a highly nonlinear space by using the topology and attribute information of the graph for symmetric normalized Laplacian transform. Then, through the feature-structured enhanced module, the node features are expanded into regular structured data, such as images and sequences, which are input to DGCGAN as positive samples, thus expanding the sample capacity. In addition, the feature-enhanced (FE) module is adopted to enhance the typicality and discriminability of node features, and to obtain richer and more representative features, which is helpful for facilitating accurate classification. Finally, additional constraints are added to the network model by introducing DCGAN, thus enhancing the robustness of the model. Through extensive empirical studies on several standard benchmarks, we find that DGCGAN outperforms state-of-the-art baselines on semi-supervised node classification and remote sensing image classification.

**Keywords:** interpolation operation; graph convolutional networks; node classification; feature-structured enhanced module

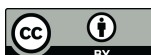

## 1. Introduction

Graphs are essential tools to model structured and relational data, such as social networks, citation networks, biological networks, the World Wide Web, and physical networks. Analyzing graph data has become one of the most important topics in the field of machine learning. Graph neural networks (GNNs), as a strong tool for deep learning, have already attracted more and more attention and achieved the most advanced performance in diverse graphical analysis tasks, including link prediction [1,2], node classification [3–6], and node clustering [7–11]. The graph convolutional network (GCN) [12] is a network framework for learning graph data structure. With the success of GCN in semi-supervised node classification, several other GCN variants have been proposed for different prediction tasks on graphs. According to the definition of convolution [13], the existing methods are divided into two categories: spatial methods and spectral methods.

The spatial method is directly defined as convolution in the vertex domain, which follows the practice of traditional CNN. In addition, some works have defined the general framework of GCN from a spatial perspective and have explained the internal mechanism of GCN. Graph neural networks (GNNs) [14] are the first model proposed to build neural networks on graphs. The mixture convolution network (MoNet) [15] focuses on the lack

of translation invariance on graphs and maps the local structure of each node to a vector of the same size by defining a mapping function, then learns the shared convolution kernel from the mapped results. The message-propagation neural network (MPNN) [16], based on information propagation aggregation between nodes, proposes a framework by defining the general form of the aggregation function. ManiReg [17] is a learning algorithm based on a new regularization form that can take advantage of the geometric properties of edge distributions. SemiEmb [18] utilizes semi-supervised regularizers to enhance deep architectures to improve their training. Feat [19] only takes the node characteristics as input and neglects the structure of the graph. The graph attention network (GAT) [20] defines an aggregation function through attention mechanism, where the adjacency matrix is used to define relevant nodes and the calculation of the correlation weight depends on the feature expression of the nodes. SGAE [21] uses a messaging mechanism to gather information from neighbors and obtain significant and independent potential representations. The open challenge faced by a spatial method is how to determine the appropriate neighborhood for a target node when defining graph convolution.

The spectral convolutional neural network (spectral CNN) [13] defines graph convolution via a convolution theorem. The original intention of modeling GCN is to use graph structure to describe the information aggregation of neighboring nodes, but spectral CNN does not satisfy locality. Henaf et al. [22] propose to use an interpolation convolution kernel with smooth constraints. This method realizes the localization of GCN. In order to make the GCN play a role in the semi-supervised learning field on graphs, Kipf et al. [12] simplify the Chebyshev network and propose a first-order GCN. Specifically, there are many models that combine GCN with generative adversarial networks [23]. For instance, GraphGAN [24] presents a framework that combines the discriminative and generative networks, in which the generator tries to approximate the true potential connectivity distribution and generates the most likely nodes connected to the target node. The existing network representation methods and related variants focus on maintaining the network topology or minimizing reconstruction errors. In comparison, GraphWGAN [25] utilizes Wasserstein distance to characterize the distance between the underlying true connectivity distribution and generated distribution in graphic representation learning. Their core idea arrives from the Nash equilibrium in the game theory. In summary, to the best of our knowledge, in semi-supervised node classification, previous works lack the number of labeled nodes, leading to low classification accuracy. The effective graph convolution methods are unable to capture the robustness of network structure.

However, when the above models and most GNNs [17–19,26–29] perform semi-supervised classification tasks, the accuracy cannot be improved due to the fewer number of samples, and the model lacks robustness. In order to address the two problems, we propose a model named deep graph convolutional generative adversarial networks (DGC-GAN) that integrates DCGAN with GCN. First, the topological structure and attribute information of the network are fully utilized for symmetric Laplace transform, and the graph data are mapped to a highly nonlinear space to obtain hidden variables and the features of each node. Then, the feature-structured enhanced block is applied to transform the node features into regular structured data as the positive samples of the discriminator. Finally, the nodes are classified by the discriminator. In addition, by introducing DCGAN, the number of samples in the model is expanded and the additional constraints are added to the network model, thus enhancing the robustness of the model expression.As illustrated in Figure 1, the adjacency matrix $A$ and the feature matrix $X$ of graph $G$ are fed into the network as inputs.

In summary, our core contributions are four-fold:

1. We use the topological structure and attribute information of the graph to perform symmetric normalized Laplace transform, map the graph data into a highly nonlinear space, and obtain the hidden variables and the features of each node.

2. The bicubic structured interpolation (BI) is used for each sample to transform the node features into regular structured data, which is sent to the discriminator as a positive sample.
3. The feature-enhanced (FE) module is introduced to selectively enhance the the typicality and discriminability of node characteristics and extract richer and more representative feature information, which is in favor of facilitating accurate classification.
4. We expand the sample size of the model and add additional constraints to the network model by incorporating DCGAN, thus enhancing the robustness of the model. We perform four benchmark datasets to confirm the validity and evaluate the properties of DGCGAN through a comprehensive comparison with state-of-the-art methods.

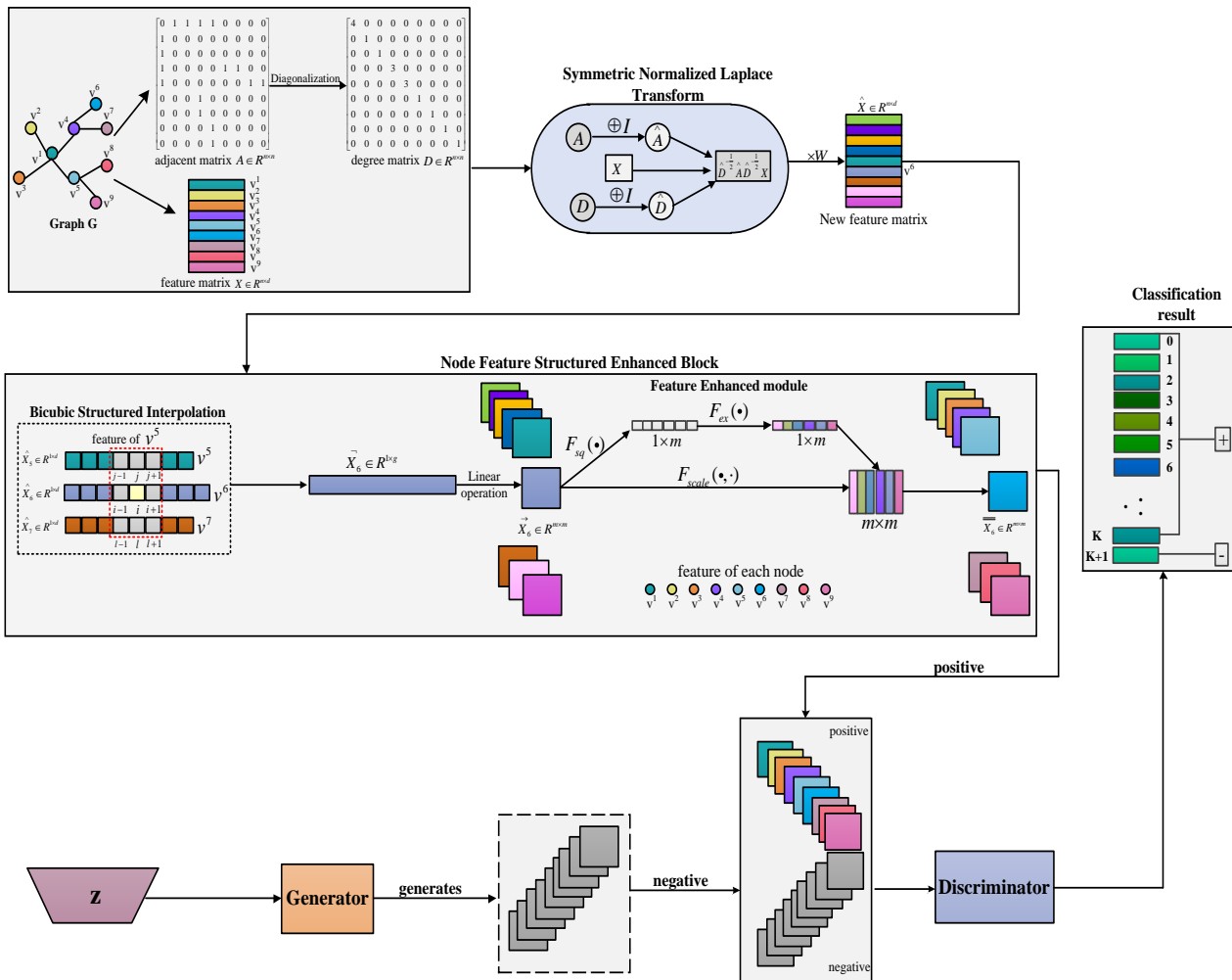

**Figure 1.** The framework of DGCGAN. The adjacency matrix *A* and the feature matrix *X* of graph *G* are fed into the network as inputs. The model is divided into four modules: the first module is symmetric normalized Laplace transform, the second module is a node feature-structured enhanced block, and the third module is DCGAN. The node feature-structured enhanced block consists of bicubic structured interpolation (BI) and feature enhanced module (FE).

## 2. Preliminary and Related Work

### 2.1. Graph Convolutional Networks

The GCN for undirected graphs as proposed by Kipf and Welling will be briefly introduced in this section. Let $G = (V, E, X)$ be a graph where $V$ is the set of $n = |V|$ nodes and $E$ is the set of $p = |E|$ edges. We denote $A \in [0,1]^{n \times n}$ as a adjacency matrix of $G$. It is a weighted adjacency matrix, and every element $A_{ij} = 1$ means that there is an edge between $v_i$ and $v_j$, otherwise $A_{ij} = 0$. The feature matrix is described as $X \in R^{n \times d}$.

The node representation after a single layer of GCN can be defined as Equation (1), where $W \in R^{d \times d'}$ is denoted as the model parameters. $I$ is the identity matrix in $\hat{A} = A + I$. The degree matrix of $A$ is defined as $D = diag(d_1, d_2, \ldots, d_n)$, where $d_i = \sum_j a_{ij}$ is the degree of node $i$. $\sigma$ is ReLU activation function. $X \in R^{n \times d}$ is the feature matrix, where $n$ is the number of nodes and $d$ is the feature dimension as in Equation (1).

$$H = \sigma((\hat{D}^{-\frac{1}{2}} \hat{A} \hat{D}^{-\frac{1}{2}}) X W) \tag{1}$$

### 2.2. Spectral GCN and Spatial GCN

Spectral GCN define convolution operation by convolution theorem. In order to make the GCN play a role in the semi-supervised learning field on graphs, Kipf et al. [12] simplify the Chebyshev Network and propose a first-order GCN. Recently, Wavelet Neural Network (GWNN) [30] has been proposed to use wavelet transform instead of Fourier transform to implement the convolution theorem. Graph Heat Kernel Network (Graph Heat) [31] uses a heat kernel function to parameterize the convolution kernel, then the model is realized a low-pass filter. The relation between GCN and PageRank is used to put forward an modified communication scheme according to personalized PageRank. This propagation program is used to construct a simple model, the personalized propagation model (PPNP) [32] is predicted by the neural network and its rapid approximate model APPNP. Simplified graph convolutional networks (SGCNs) [33] reduces the extra complexity of GCNs by repeatedly eliminating the nonlinearity between GCN layers, folding the resulting function into a linear transformation, and proving that SGC is equivalent to a fixed low-pass filter plus a linear classifier.

Spatial GCN immediately establishes the convolution operations based on the connectivity of each node, which is more analogous to the convolution in the conventional CNN. Based on applying the model on large-scale networks, Hamilton et al. [34] propose a graph sampling aggregation network (GraphSAGE) which randomly samples neighboring nodes so that the neighboring nodes of each node are fewer than the given number of samples. DCNN [35] uses the K-hop transition probability obtained after randomly walking to define the weight between nodes. The confidence-based graph convolutional network (ConfGCN) [36] believes that a node is a label with a certain degree of confidence. ConfGCN learns a confidence function for each node and applies it to calculate the correlation of the node to modify the aggregation function. The hypergraph convolutional network (HGNN) [37] realizes that the correlation between nodes is mutually influenced by a group of nodes. A group of nodes can build the correlation of nodes within the group. Relational graph neural networks (R-GCNs) [38] divide the original network into different sub-networks according to the direction of the graph of edges and the label type on the edges, then independently aggregate neighbor features on each sub-graph. The jumping knowledge network [39] uses jump connections and an attention mechanism to select the appropriate propagation range for each node. Zhang et al. [40] propose a traffic data completion model based on GCN to complete the lacking values based on deep learning. There are also some GNN tasks based on graph similarity calculation tasks, including [41–44]. ML-GCN [45] is proposed as a new multi-label GCN, which is utilized to learn the node representation of multi-label networks. However, most of these models are restricted by poor robustness when performing classification tasks.

### 2.3. Deep Convolutional Generative Adversarial Network

DCGAN [46] is a relatively good improvement after GAN. It is main improvement is the network structure. The network structure of DCGAN is still widely used and can greatly improve the stability of GAN training and the quality of generated results.

Compared with the original GAN, DCGAN almost completely only uses convolutional layers without fully connected layers. The discriminator is almost symmetrical to the generator. The entire network does not have any pooling layers or up-sampling layers. In fact, it utilizes a fraction-stride convolution instead of up-sampling to increase the stability of training.

### 2.4. Semi-Supervised Learning on Graphs

For the semi-supervised classifying task, $m$ nodes have already detected $Y^L$ and the labels $Y^U$ of the residual $n - m$ nodes are missing. Its goal is to acquire a prediction function $g : G, X, Y^L \to Y^U$ to conclude the absent labels $Y^U$ for unlabeled nodes, a label vector $Y_i \in Y \in \{0,1\}^{n \times c}$ in which $c$ indicates the quantity of classes.

As a method of data-mining analysis, node classification is widely used in social networks. The nodes in the social network can represent a wealth of extra data such as text, images, and their own attribute values. By analyzing these characteristic information, the nodes in the network can be classified and labeled. In the node-classification task, nodes cannot be classified as independent individuals because of the connection relationship between nodes. Node classification is an significant branch of graph data mining. Through analyzing the network structure and relationship between nodes and edges during semi-supervised learning, labeled nodes are used to classify unlabeled nodes more accurately, avoiding the trouble of manual labeling and the high cost of additional calculation.

## 3. Deep Graph Convolutional Generative Adversarial Networks

### 3.1. The Overall Architecture

The DGCGAN framework is illustrated in Figure 1. DGCGAN comprises three parts: the symmetrical normalized Laplace transformation, node feature-structured enhanced block, and DCGAN. The node feature-structured enhanced block consists of interpolation operation and FE. As shown in Figure 1, taking node $v_6$ as an example, through the interpolation operation, we obtain the feature of node $v_6$, $\overleftarrow{X_6} \in R^{1 \times g}$. After the linear operation, we obtain a new feature representation of node $v_6$, $\overrightarrow{X_6} \in R^{m \times m}$. Through FE operation, the feature of node $v_6$, $\overline{\overline{X_6}} \in R^{m \times m}$ is obtained, which is a positive sample to the discriminator. The random noise $z$ obeying Gaussian distribution ($z \sim N(0,1)$) is used as the input of the generator, then the generated node features are sent to the discriminator as negative samples. The task of the discriminator is not only to discriminate the negative samples produced by the generator, but also to classify the positive samples, including the $K$ categories in total.

Firstly, the features of each node are obtained by using the topological structure and attribute information of the graph to perform symmetric normalized Laplace transform in Section 3.2. Then, by using the node feature-structured enhanced block, we transform the node features into regular structured data, and send them as positive samples to the discriminator as input in Section 3.3. Thirdly, we selectively enhance the typicality and distinguishability of node features by adopting the FE module. Finally, by incorporating DCGAN, we expand the sample size of the model and add additional constraints to the network model, thus enhancing the robustness of the model in Section 3.4. The specific algorithm flow of DGCGAN model is shown in Algorithm 1.

---

**Algorithm 1:** The framework of DGCGAN.

---

**Input:** Graph structure $G$ containing adjacency matrix
$A \in R^{n \times n}$ and feature matrix $X \in R^{n \times d}$
**Output:** $K + 1$ dimensional vector $R$
1  **for** $i = 1, 2, \ldots, n$ **do** ($n$ is the number of nodes)
   Initialize the $\tilde{A} = A + I$, $\tilde{D}_{ii} = \sum_{j=1}^{n} \tilde{A}_{ij}$
2    Symmetric normalized Laplace transformation (3)
     to calculate the feature matrix $\hat{X} \in R^{n \times d}$
3    According to the $i$-th node feature $\hat{X}_i \in R^{1 \times d}$ to obtain
     $\overrightarrow{X_i} \in R^{1 \times g}$ by bicubic structured interpolation, obtaining
     $\tilde{X}_i \in R^{m \times m}$ by reshape operation.
4    Feature squeeze and feature excitation operation: using Equations (6)–(8)
     to calculate
     $F_{sq}(\tilde{X}_i) = Z_i \quad F_{ex}(Z_i) = S_i \quad F_{scale}(\tilde{X}_i, Z_i) = \overline{\overline{X_i}}$
5    Input random noise to the generator to
     obtain $G(z)$
6    **for** $j = 1$ to $T$ **do** ($T$ is the total number of iterations)
        Taking two positive samples from the set
        $G(z_1), G(z_2), \ldots, G(z_n)$ and taking two negative
        samples from the set $\left\{ \overline{\overline{X_1}}, \overline{\overline{X_2}}, ..., \overline{\overline{X_n}} \right\}$
7       discriminator $D(v) \leftarrow \overline{\overline{X_i}}, \overline{\overline{X_i + 1}}$
        generator $G(z) \leftarrow G(z_i), G(z_{i+1})$
8       According to Equation (11), calculate unsupervised
        loss function $L_{unsupervised}$
9       The discriminator performs supervised
        learning to obtain $K + 1$ dimensional vector
        $R$ and updates the full supervised loss function
        $L_{supervised}$
10      **end for**
11  **end for**
12  **return** the classification results $R$

---

### 3.2. Symmetric Normalized Laplace Transform

For the graph $G = (A, X)$, where $A$ is the adjacency matrix and $X$ is the feature matrix, $D = diag(d_1, d_2, \ldots, d_n)$ is a degree matrix with $d_i = \sum_{j=1}^{n} A_{ij}$. $L = D - A$ is defined as the Laplace matrix. Laplacian smoothing is performed on the graph, according to Equation (2),

$$L = (I - \gamma \hat{D}^{-1} \hat{L}) X \tag{2}$$

where $\hat{L} = \hat{D} - \hat{A}$, $\hat{D} = D + I$, $\hat{A} = A + I$, $0 < \gamma \leq 1$ is a hyper parameter that is used to adjust the weight of target node and surrounding nodes.

Let $\gamma = 1$, $L = \hat{D}^{-1} \hat{A} X$ be obtained after Laplacian smoothing (using Equation (2)). The spatial perspective of node $v_i$ has a larger degree, and the eigenvalues after aggregation are far larger than those of the node $v_j$ with a smaller degree. Therefore, before the node updates itself, it is necessary to normalize the information transmitted by neighboring nodes to eliminate the problem. However, normalization only considers the degree of the node itself, but does not consider the degree of the neighbor nodes. This will make the neighbor node with the larger degree obtain the smaller weight value, which may lead to the vanishing gradient or the exploding gradient in the network-training process. Finally, we use the symmetrically normalized Laplace operator $\hat{D}^{-\frac{1}{2}} \hat{L} \hat{D}^{-\frac{1}{2}}$ to replace $\hat{D}^{-1} \hat{L}$. Consequently, Equation (2) is equivalent to $\hat{D}^{-\frac{1}{2}} \hat{A} \hat{D}^{-\frac{1}{2}} X$. In Figure 1, the updated nodes are normalized by symmetric normalized Laplace transformation, and then undergo a layer

of GCN transform in Equation (3), where $W \in R^{d \times d'}$. Therefore, the essence of GCN is Laplacian smoothing over the network. The feature matrix $\hat{X}$ contains not only the topology information of the graph $G$, but also the node feature information. Experiments show that the classification effect of symmetric normalized Laplacian operator is better than that of the normalized Laplacian operator in [12].

$$\hat{X} = \sigma(\hat{D}^{-\frac{1}{2}} \hat{A} \hat{D}^{-\frac{1}{2}} XW) \tag{3}$$

We take the adjacency matrix $A \in R^{n \times n}$ and the feature matrix $X \in R^{n \times d}$ as the inputs of the network, where $n$ is the number of nodes and $d$ is denoted the feature dimension of nodes.

### 3.3. Node Feature-Structured Enhanced Block

The node feature-structured enhanced block is divided into two parts: the bicubic structured interpolation (BI) and feature-enhanced (FE) modules.

#### 3.3.1. Bicubic Structured Interpolation

The interpolation method we use in this paper is bicubic structured interpolation. This method uses the gray values of 9 points to be sampled for cubic interpolation, which takes into account not only the gray values of four directly adjacent points, but also the change rate of gray values between adjacent points. Assume that the size of the original image A and the scaled target image B are, respectively, $m \times n$ and $M \times N$. If the pixel value at $B(X, Y)$ is required, we can first obtain the corresponding position $(x, y) = (X \times \frac{m}{M}, Y \times \frac{N}{n})$ of $B(X, Y)$ in the image $A$. At this time, there are decimal parts in $x$ and $y$, so we can find the nearest 9 pixels by this decimal coordinate and use the selected basis function to find the corresponding weight of each pixel; finally, we receive the pixel value at $B(X, Y)$.

By constructing a bicubic structured (BS) function (see Equation (4)), we obtain the nine weights $W_j (j = 1, 2, 3, \ldots, 9)$ corresponding to the three dimensions of the three target nodes by obtaining the parameter $x$ in the BS function, where $x$ represents the $i$-th dimension of the target node $v$ and $a = -0.5$. In Figure 1, the value corresponding to the $i$-th dimension of the target node $v_6$ is $BS(v_i)$ (see Equation (5)).

$$W(x) = \begin{cases} (a+2)|x|^3 - (a+3)|x|^2 + 1 & |x| \leq 1 \\ a|x|^3 - 5a|x|^2 + 8a|x| - 4a & 1 < |x| < 2 \\ 0 & otherwise \end{cases} \tag{4}$$

$$BS(v_i) = \sum_{j=1}^{9} a_j W_j \tag{5}$$

In Figure 1, we use $\hat{X}_i \in R^{1 \times d} (i = 1, 2, \ldots, n)$ to represent the feature of the $i$-th node. The values corresponding to the $j-1$, $j$, $j+1$ dimensions of node $v_5$ are $a_1$, $a_2$, and $a_3$, and the corresponding weights are $W_1$, $W_2$, and $W_3$. The values of $i-1$, $i$, and $i+1$ dimensions corresponding to node $v_6$ are $a_4$, $a_5$, and $a_6$, and the corresponding weights are $W_4$, $W_5$, and $W_6$. The values corresponding to the $l-1$, $l$, $l+1$ dimensions of node $v_7$ are $a_7$, $a_8$, and $a_9$, and the corresponding weights are $W_7$, $W_8$, and $W_9$. After BI operation, the feature of the $i$-th node becomes $\overrightarrow{X}_i \in R^{1 \times g} (i = 1, 2, \ldots, n)$. Among them, $g$ and $d$ are both integers and $g$ is greater than $d$. We convert $\overrightarrow{X}_i \in R^{1 \times g}$ into $\tilde{X}_i \in R^{m \times m}$ by linear operation, where $m = \lfloor \sqrt{g} \rfloor$.

We perform a bicubic structured interpolation on each node to work out the feature of each node. By the bicubic structured interpolation method, the feature dimension of nodes is extended, thus providing rich unstructured information to enhance the performance of graph embedding, and then graph embedding is further used to obtain more typical and indistinguishable node features through the feature-enhanced module.

### 3.3.2. Feature-Enhanced Module

FE adaptively recalibrates the characteristics of channels by explicitly modelling the interdependencies among the channels. Its core idea is tantamount to learn feature weights through the network in order to achieve better results in the manner of an active feature map with significant weight, or an invalid or ineffective feature map with a low weight. In Figure 2, the FE module is mainly composed of three parts: the feature squeeze operation $F_{sq}(\cdot)$, feature excitation operation $F_{ex}(\cdot)$, and feature scale operation $F_{scale}(\cdot, \cdot)$.

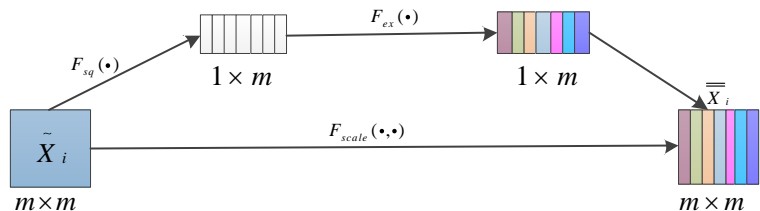

**Figure 2.** The workflow of FE module. Using the feature $\tilde{X}_i \in R^{m \times m}$ of the *i*-th node for input, and the feature dimensions as channels. By the feature squeeze operation $F_{sq}(\cdot)$ and the feature excitation operation $F_{ex}(\cdot)$, a vector $T$ of $1 \times m$ is obtained and multiplied by $\tilde{X}_i \in R^{m \times m}$ by $F_{scale}(\cdot, \cdot)$ manipulation. the new feature of the *i*-th node $\overline{\overline{X}}_i$ is obtained and is put into the discriminator.

Feature squeeze operation. In order to solve the problem of global information embedding utilizing channel correlation, we first consider the signal of each channel of the output characteristics. Each feature dimension of a node is regarded as a channel and the feature of each node is obtained by using global average pooling. Let $\tilde{X}_j \in R^{m \times m}$ be the feature of the *j*-th node. After feature squeeze operation (see Equation (6)), the feature of the *j*-th node is transformed into $Z_j \in R^{1 \times m}$. We compress the $m \times m$ dimension to the $1 \times m$ dimension, thus obtaining the global receiving field information, where $m$ is the dimensionality of the properties of the *j*-th node and $n$ is the number of total nodes.

$$Z_j = F_{sq}(\tilde{X}_j) = \frac{1}{n} \sum_{i=1}^{n} \tilde{X}_i \qquad (6)$$

Feature excitation operation. To take advantage of the information gathered in the compression operation, we trace it using the feature excitation operation, which is designed to capture the channel dependencies completely. It is similar to the gated mechanism in the recurrent neural network, giving every part a relative weight, which is analogous to the attention mechanism. The feature excitation operation is first performed after a full connection operation on the node embedding generated by the feature squeeze operation, then non-linear mapping (ReLU function) is performed. After that, a full connection operation and a sigmoid are performed to obtain the weight of each feature dimension of the node in Equation (7), where $W_1 \in R^{m \times m}$, $W_2 \in R^{1 \times m}$, $Z_j \in R^{1 \times m}$ is the node embedding of the *j*-th node, $S_j \in R^{1 \times m}$ is the feature of the *j*-th node obtained.

$$S_j = F_{ex}(Z_j, W) = \sigma(g(Z_j, W)) = \sigma(W_2 \delta(W_1 Z_j)) \qquad (7)$$

Feature scale operation. We use the weight of the feature exciting manipulation output as the significance of different features selected, and then multiply every feature dimension by the embedding $\tilde{X}_j$ of the previous feature to assign a value to achieve the re-scaling of the initial feature dimension. Figure 2 illustrates the importance of the export weight of feature excitation operating as the feature dimension when the feature is selected, and then the former feature is weighted by multiplication. Finally, the primary features are recalibrated in size. See Equation (8) for the detailed procedures, where $\overline{\overline{X}}_j \in R^{m \times m}$.

$$F_{scale}(\tilde{X}_j, S_j) = S_j \tilde{X}_j = \overline{\overline{X}}_j \qquad (8)$$

### 3.4. Deep Convolutional Generative Adversarial Networks

#### 3.4.1. Generator

The traditional GAN model is composed of two parts of the network: namely, the discriminator network that performs authenticity identification and the generator network that generates false samples. The generator generated some negative samples with pseudo-labels, thus expanding the number of samples.

Under unsupervised learning, the DCGAN actually performs the binary classification tasks of authenticity identification. Its purpose is to discern between the true samples and fake samples generated by the generator network. The proposed model constructs a classification network model based on the discriminator network. The classification network performs supervised learning on labeled data and constitutes a semi-supervised classification model together with unsupervised learning (see Figure 3 for details). Assuming that there are $K$ categories of the data samples for network training, the semi-supervised classification network performs a $K + 1$ category-classification task and the $K + 1$-th category is the false sample category generated by the generator network. In the deep convolutional graph convolutional generative adversarial network (DGCGAN), the generator G consists of four layers of deconvolutions. The generator network receives a random noise $z$ and continuously optimizes the distribution of authentic samples through a penalty term of the discriminator network. See Figure 3 for DCGAN; a 256 dimensional uniform distribution $z$ is projected to a small spatial extent convolutional representation with many feature maps. Then, the upsampling is performed through the upsampling mechanism to generate a node feature map, and finally, all layers except for the output use ReLU activation function, which uses Tanh function. The batch norm is used in all layers.

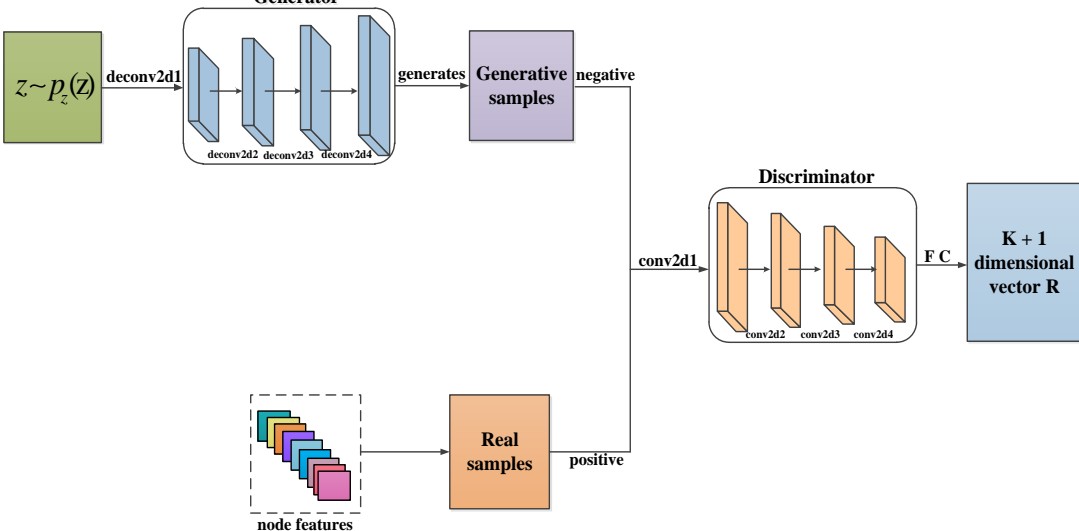

**Figure 3.** The flow chart of deep convolutional generative adversarial networks. The generator includes four deconvolutional layers and the discriminator includes four convolutional layers.

#### 3.4.2. Discriminator

The discriminator $D(v)$ is no longer a simple true or false classifier. Assuming that the input data have $K$ classes, $D$ is currently a classifier of $K + 1$ classes; the extra class is to determine whether the input is an image generated by generator $G(z)$. The discriminator $D(v)$ consists of four convolutional layers and uses two-dimensional convolution to downsample the feature map of each node, then pass it to the fully connected layer for classification. LeakyReLU activation is used after all layers of discriminator. The positive and negative samples are judged first, then the positive samples are classified as shown in Figure 3.

The generator $G(z)$ and discriminator $D(v)$ act as two opponents: the generator $G(z)$ would tries to fit $X_{true}(v_i)$ (real nodes' feature map) perfectly and generate feature maps similar to real nodes to deceive the discriminator, while the discriminator $D(v)$ tries to detect whether the feature map of these nodes are real feature maps of nodes or the ones generated by its counterpart generator $G(z)$.

### 3.5. Loss Function

If the input data is a positive sample, the target function that is would be maximized $logP(y \in \{1, 2, \ldots, K\}|v)$. The loss function designing of the discriminator $D(v)$ includes two components in which one is the supervision learning loss function and the another is the unsupervised learning loss function, as described in Equation (9).

$$L = L_{dis} + L_c \tag{9}$$

The supervised part of the loss functions is defined as Equation (10); for the unsupervised part of the task, the only thing that needs to be done is to distinguish whether a sample is positive or negative. The unsupervised part of the loss functions is expressed by the Equation (12), where $P$ represents the probability of discriminating negative samples.

$$L_c = - \mathbf{E}_{v,y \sim p_{true}(v,y)} \log P \\ (y|v, y < K + 1) \tag{10}$$

$$D(v) = 1 - P(y = K + 1|v) \tag{11}$$

The unsupervised loss function part is expressed as Equation (12).

$$L_{dis} = - \left\{ \mathbf{E}_{v \sim p_{true}(v)} log D(v) + \\ \mathbf{E}_{z \sim noise} log(1 - D(G(z))) \right\} \tag{12}$$

The final target function that needs to be optimized for the DGCGAN can be expressed as Equation (13).

$$\min_G \max_D \mathbf{E}_{v \sim P_{true}(v)}[log D(v)] + \mathbf{E}_{z \sim noise} \\ [log(1 - D(G(z)))] \tag{13}$$

### 3.6. Complexity Analysis

The time complexity of DGCGAN is $O(|E|dd' + |V|m)$, where $d$ is denoted as the feature dimension of nodes and $d'$ is denoted as the dimension of output features. $m$ is the output dimension of the node feature-structured enhanced block. $|E|$ and $|V|$ are the numbers of nodes and edges in the graph, respectively. The complexity is comparable to that of other GNN methods such as GCN and GAT. The time complexity of DGCGAN is mainly determined by the following two parts, which is $O(|E|dd')$ for GCN and $O(|V|m)$ for the node feature-structured enhanced block.

## 4. Experiments and Analysis

### 4.1. Datasets

In order to fully demonstrate the validity of model DGCGAN in semi-supervised node classification, we conduct an evaluation on various semi-supervised classification benchmarks. According to the experimental settings [12,47], we evaluate it on Cora, Citeseer, Pubmed, and ogbn-arxiv datasets [47]. The detailed information of these datasets and their applications in our research are described below. The dataset attributes are summarized in Table 1.

**Table 1.** Details of the datasets utilized in our experiments.

| Datasets | Nodes | Edges | Classes | Features | Labeling Rate |
|----------|-------|-------|---------|----------|---------------|
| Cora | 2708 | 5429 | 7 | 1433 | 0.052 |
| Citeseer | 3327 | 4732 | 6 | 3703 | 0.036 |
| Pubmed | 19,717 | 44,338 | 3 | 500 | 0.003 |
| ogbn-arxiv | 169,343 | 1,166,243 | 40 | 128 | 0.537 |

**Cora**. This dataset consists of 2708 nodes and 5429 edges. All nodes have 1433-dimensional feature descriptions, and all nodes are divided into seven categories.

**Citeseer**. This dataset contains 3327 nodes and 4732 edges. All nodes belong to six categories. Every node has a dimensional feature description of 1433.

**Pubmed**. This dataset is a biomedical paper search and abstract data set. The data contains $19,717$ nodes and 44,338 edges. All nodes are divided into three categories and each node has a 500-dimensional feature descriptor.

**Ogbn-arxiv**. The dataset includes $169,343$ nodes and $1,166,243$ edges. All nodes are split into forty categories. Each node obtains a dimension feature descriptor of 128.

*4.2. Baselines*

To evaluate the model DGCGAN, we compare it with the following baselines: LP [48], DeepWalk [49], GCN [12], G-GCN [50], GPNN [51], GAT [20], SGCN [33], ConfGCN [36], APPNP [32], HGCN [52], HGNN [53], GSNN-M [54], IDGL [55], GraphSAINT [56], and HAT [57].The commonly used evaluation indexes of classification algorithms include accuracy, precision, recall, and F1 score. We use accuracy in this experiment, which is a commonly used evaluation index. The precision of specific classification is indicated in Table 2.

**Table 2.** Performance comparison of different methods for semi-supervised node classification accuracy (percentage of standard deviation) on multi-benchmark datasets. DGCGAN is consistently better on all datasets. OOM indicates that the memory is exceeded.

| Method | Cora | Citeseer | Pubmed | Ogbn-Arxiv |
|--------|------|----------|--------|------------|
| LP | $67.9 \pm 0.1$ | $45.1 \pm 0.2$ | $62.7 \pm 0.3$ | OOM |
| MLP | $56.1 \pm 1.6$ | $56.7 \pm 1.7$ | $71.4 \pm 0.0$ | $54.7 \pm 0.1$ |
| GRCN | $67.4 \pm 0.3$ | $67.3 \pm 0.8$ | $67.3 \pm 0.3$ | OOM |
| DeepWalk | $67.0 \pm 1.0$ | $47.8 \pm 1.6$ | $73.9 \pm 0.8$ | $55.6 \pm 0.3$ |
| GCN | $81.4 \pm 0.1$ | $70.1 \pm 0.2$ | $78.7 \pm 0.3$ | $69.9 \pm 0.1$ |
| G-GCN | $81.2 \pm 0.4$ | $69.6 \pm 0.5$ | $77.0 \pm 0.3$ | $68.2 \pm 0.3$ |
| GPNN | $79.0 \pm 1.7$ | $68.1 \pm 1.8$ | $73.6 \pm 0.5$ | $63.5 \pm 0.4$ |
| GAT | $83.0 \pm 0.7$ | $72.5 \pm 0.7$ | $78.8 \pm 0.2$ | OOM |
| SGCN | $81.0 \pm 0.0$ | $71.9 \pm 0.1$ | $78.9 \pm 0.0$ | $68.3 \pm 0.6$ |
| ConfGCN | $82.0 \pm 0.3$ | $72.7 \pm 0.8$ | $79.5 \pm 0.5$ | $68.9 \pm 0.2$ |
| GSNN-M | $83.9 \pm 0.4$ | $72.2 \pm 0.4$ | $79.1 \pm 0.3$ | $69.8 \pm 0.2$ |
| IDGL | $70.9 \pm 0.6$ | $68.2 \pm 0.6$ | $72.3 \pm 0.4$ | OOM |
| HGCN | $80.5 \pm 0.6$ | $71.9 \pm 0.4$ | $79.3 \pm 0.5$ | $55.6 \pm 0.2$ |
| HAT | $83.6 \pm 0.5$ | $72.2 \pm 0.6$ | $79.6 \pm 0.5$ | OOM |
| APPNP | $81.2 \pm 0.2$ | $70.5 \pm 0.1$ | $79.8 \pm 0.1$ | $69.1 \pm 0.0$ |
| DGCGAN | $84.5 \pm 0.1$ | $73.3 \pm 0.1$ | $82.2 \pm 0.2$ | $71.2 \pm 0.1$ |

LP [48] is an recursive label propagating algorithm which distributes a node's label to its neighboring unlabeled nodes based on their proximity. DeepWalk [49] treats random walks in the graph as equivalent sentences to learn node features. GCN [12] is a variable of the convolutional neural network which is applied to the semi-supervised learning of graphic data structure. G-GCN [50] proposes a GCN-based semantic role labeling (SRL) model which can fuse syntax information in the sequence model.GPNN [51] can be run

on a super-large graph. The core idea is to use a modified multi-seed flood fill to quickly divide the large graph, thereby spreading local information between nodes of the subgraph and spreading global information between subgraph alternatives. GAT [20] is a method based on graph attention, which provides different weights for different nodes by allowing nodes to pay attention to their neighborhoods. SGCN [33] transforms the nonlinear GCN into a simple linear model SGCN and reduces the extra complexity of GCNs by repeatedly eliminating the nonlinearity between the GCN layers and folds the resulting function into a linear transformation. ConfGCN [36] aims to predict the marking scores of the remaining nodes in the graph by marking a few nodes as seeds and then using the graph structure. APPNP [32] combines the graph convolution neural network with personalized page ranking, and constructs a simple model with PageRank propagation to better propagate neighborhood node information. HGCN [52] aggregates the expressive ability of GCN and hyperbolic geometry to learn the node representation in scale-free graphs or hierarchies. HGNN [53] consists of an encoder based on a heterogeneous graph and an emotional personality perception decoder. GSNN-M [54] uses confidence estimation in the context of GCN to estimate label scores. The model adopts these estimated confidence levels to confirm the effect of a node on another during neighbor aggregation, thus obtaining anisotropic capabilities. IDGL [55] proposes an end-to-end graph-learning framework for combining the iterative learning graph structure and graph embedding. GraphSAINT [56] designs a lightweight sampling algorithm to reduce sampling variance by quantifying the influence of node neighbors. HAT [57] exploits the graph attention network to learn the robust node representation of graphs in hyperbolic spaces.

*4.3. Performance Comparison*

4.3.1. Node Classification

Node classification is a fundamental task extensively used to evaluate the effect of embedding. Table 2 shows the classification accuracy of DGCGAN and other baselines in semi-supervised scenarios. LP, GCN, GPNN, GAT, ConfGCN, GSNN-M, and HAT are semi-supervised models that can be used directly to classify nodes.For DeepWalk, GGNN, and IDGL, etc., we adopt the KNN classifier with $k = 5$ to perform node classification. The results are summarized in Table 2. It is clear that DGCGAN performs best in most cases. In addition, we can realize that GNN-based methods generally perform better than other baselines (i.e., LP, MLP, DeepWalk, and GRCN) since graph topology and node features are combined in their models. Compared with GCN and HGCN, GAT and HAT have achieved better results in most cases, which is due to the multi-head attention mechanism of GAT and HAT. Furthermore, the results on ogbn-arxiv illustrates the scalable performance of DGCGAN.

4.3.2. Node Clustering

As for the node-clustering task, we estimate the property of the method by analyzing the quality of the learned representation. Specifically, for the GNN-based method, we train our framework to represent the test nodes of the hidden layer by obtaining features. In order to make a fair comparison, we set the dimension of features to 16, use K-means to execute node clustering, and the number of clusters is set to the number of labels. NMI (normalized mutual information) is often utilized in clustering algorithms to measure the similarity between two clustering results (usually we compare the similarity between clustering results and real labels). The difference between NMI and another clustering index ACC is that the value of NMI will not be affected by the arrangement of family labels. We use the average results of normalized mutual information (NMI) of 10 runs with random weight initialization. In Table 3, we show the results of node clustering. As shown in Table 3, we can see that the performance of DGCGAN is superior to all baselines all the time, which shows that the performance improvement of DGCGAN is really conducive to clustering tasks.

**Table 3.** The results of the node-clustering task in the structural optimization scenario (NMI metric in percentage).

| Method | Cora | Citeseer | Pubmed |
|---|---|---|---|
| DeepWalk | 40.4 | 17.9 | 23.1 |
| GCN | 51.7 | 42.4 | 30.5 |
| GraphSAINT | 50.1 | 38.7 | 32.5 |
| GAT | 56.7 | 42.7 | 35.9 |
| Node2vec | 41.5 | 20.9 | 28.6 |
| HGCN | 57.2 | 42.1 | 37.2 |
| HGNN | 55.3 | 41.3 | 38.7 |
| DGCGAN | 59.2 | 44.1 | 40.2 |

### 4.3.3. Remote Sensing Scene Classification

In addition to the citation network dataset, we also classified it on NWPU-RESISC45 (NWPU) dataset. It is made up of 45 scene class, with a total of 31,500 remote sensing scenes images. Each scene class contains 700 scene images with $256 \times 256$ pixels. This dataset is one of the largest in the number of scene categories and the total number of scene images. During the experiment, 10% and 20% are randomly selected images of each scene category as the training group, and the rest are divided into test groups (20/80 NWPU). Firstly, we construct an adjacency matrix with topological structure through the dataset NWPU. The specific operation is to regard each picture as a node and the label of the node as a criterion for judging whether there is an edge with other nodes. Then, the pixel matrix of the picture is taken as the feature matrix of the graph. Finally, they are jointly transported to DGCGAN.

The comparison results between the proposed method and the prior state-of-the-art method are listed in Table 4. As can be seen from Table 4, the OA of the proposed method is only moderately lower than that of the best method, DGCGAN. For the training test ratio of 2:8, the average OA of the proposed method is 2.43% and 4.15% lower than that of the optimal method DGCGAN. In this paper, most comparison methods with high classification performance employ pretraining methods. Compared with the CNN model trained from scratch, the pretrained CNN model can achieve significantly better classification performance. To visualize the final convolution output of DGCGAN, we select representative images from NWPU dataset and input them into the network, respectively. As shown in Figure 4, the gradient-weighted class activation mapping (Grad-CAM) method is used for visualization. The proposed method DGCGAN utilizes the node feature-structured enhanced block to extract richer and more representative feature information, which is helpful to solve the problems of intra-class differences and inter-class similarities in remote sensing scene image classification, thus contributing to correct classification.

**Table 4.** Comparison results of OA on the NWPU test set.

| Basic Network | Methods | OA (20/80) (%) |
|---|---|---|
| AlexNet | MSCP | $85.58 \pm 0.16$ |
| VGG16 | MSCP | $88.93 \pm 0.14$ |
| AlexNet | SCCov | $87.30 \pm 0.23$ |
| GCN | DGCGAN | $89.73 \pm 0.17$ |

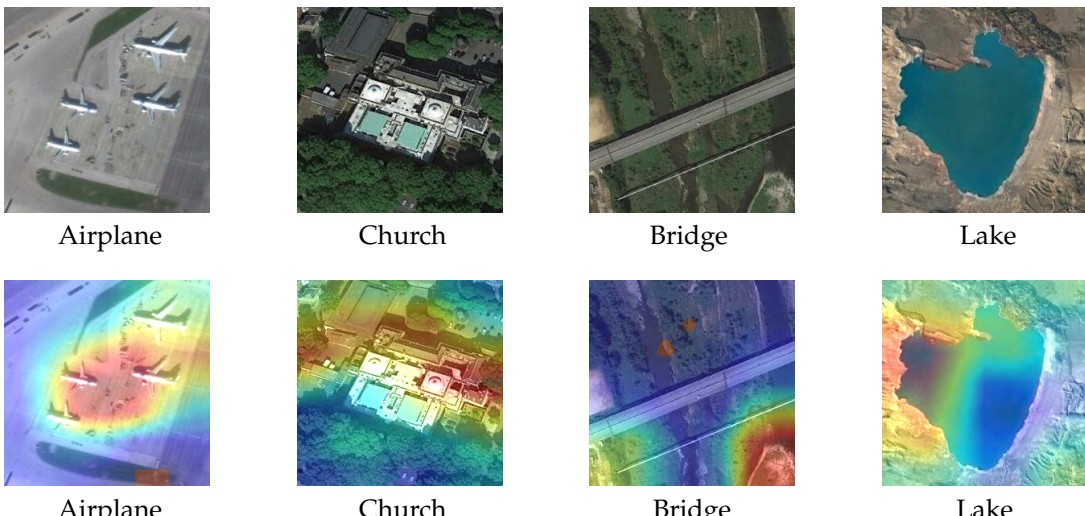

| Airplane | Church | Bridge | Lake |

| Airplane | Church | Bridge | Lake |

**Figure 4.** The visualization results of DGCGAN on the NWPU dataset.

### 4.4. Ablation Study

In order to verify the effectiveness of the proposed DGCGAN model, we conduct an ablation study in this section.

#### 4.4.1. The Effectiveness of Symmetric Normalized Laplace Transform

The Laplace matrix is a matrix representation of a graph. The Laplacian mapping directly decreases the dimension to a low-dimensional space while maintaining the similarity of the original manifold data. The spatial significance of normalizing Laplace transform is that each node sends the same amount of information through the edge, and the more edges there are, the less information each edge sends. In other words, the greater the degree of neighboring nodes, the smaller the weight assigned. In order to avoid this situation, we normalize the Laplace matrix. In Table 5, we compare three variants of the propagation model on the the citation network datasets. Specifically, we compare the performance of MLP, Laplace transform (LP), and symmetric normalized Laplace transform (SNLP) on the datasets Cora, Citeseer, and Pubmed. Generally speaking, we find that SNLP has the best performance among all propagation models.

**Table 5.** The comparison results of propagation models.

| Description | Propagation Model | Cora | Citeseer | Pubmed |
|---|---|---|---|---|
| MLP | XW | 55.1 | 46.5 | 71.4 |
| LT | (D-A)XW | 80.5 | 68.7 | 77.8 |
| SNLT | $\hat{D}^{-\frac{1}{2}}\hat{A}\hat{D}^{-\frac{1}{2}}XW$ | 81.5 | 70.3 | 79.0 |

#### 4.4.2. The Effectiveness of BI and SE

First, we combine DCGAN and GCN to expand the sample size. Secondly, the addition of bicubic structured interpolation converts the node features into regular graph data, where SE stands for FE. Finally, the addition of the feature squeeze and feature excitation module strengthens the typicality and discriminability of node features so that the robustness of the model is enhanced. The results are shown in Figure 5.

The Figure 5 presents that the classification accuracy of GCN in the three benchmark graph datasets (Cora,Citeseer, and Pubmed), respectively, are 81.5%, 70.3%, and 79%. First, we fuse DCGAN and GCN to expand the number of samples and enhance the robustness of the model. We can obtain the classifying accuracy of three benchmark datasets: 84.1%, 72.8%, and 81.4%. Second, we add the bicubic structured interpolation to the model

above and acquire the precision as follows: 84.3%, 73.0%, and 81.8%, respectively. Third, combining feature squeezing and feature excitation operating into the above model, the classifying accuracy is obtained, which are 84.5%, 73.3%, and 82.3%, respectively.

In addition, in the bicubic structured interpolation operation, we find that the feature dimension affects the classification result. We take Cora and Citeseer as examples. In Cora, the feature dimension is $d = 1433$. We take the parameter $g$ as 1369, 1444, and 1521 and the classification accuracy results are 84.1%, 84.5%, and 78.6%, respectively. In Citeseer, feature dimension is $d = 3703$; we take $g$ as 3600, 3721, and 3844, respectively, and the corresponding classification results are 72.9%, 73.3%, and 72.8% (Figure 6). We find that the closer the parameter $g$ and the feature dimension $d$ are, the better the classification results will be. The corresponding classification accuracy is shown in Table 6.

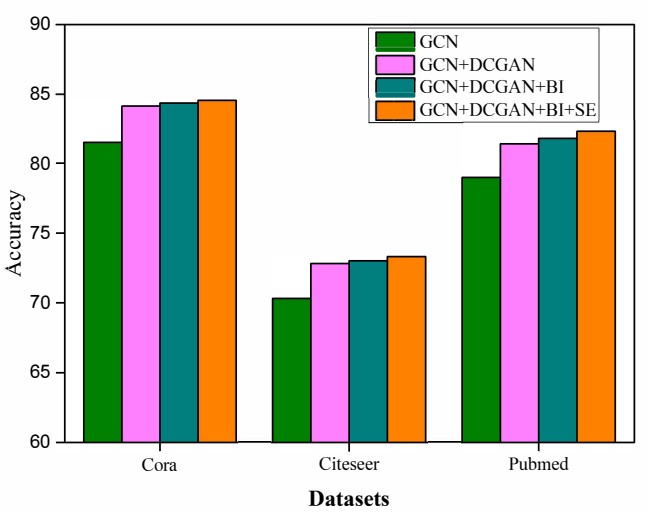

**Figure 5.** The accuracy results of model GCN, combining GCN and DCGAN, adding BI operation to the model, adding SE (FE) module to the model on node-classification task.

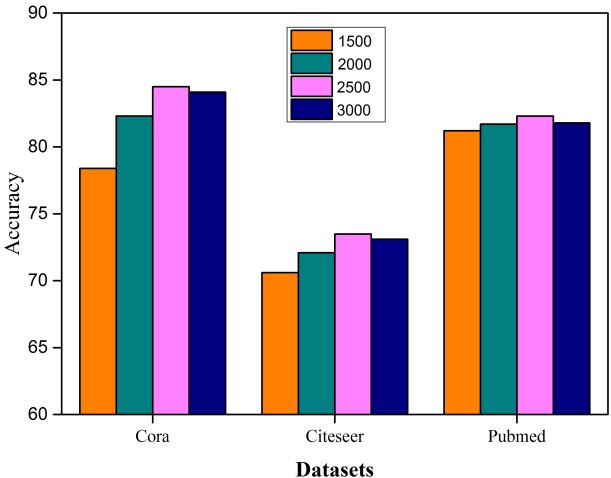

**Figure 6.** Compare the performance of different sample augment on the diverse datasets Cora, Citeseer, and Pubmed.

### 4.4.3. Sample Augment Analysis

We find that the number of samples of the generator will affect the classification results. To verify the idea, we input 1500, 2000, 2500, and 3000 samples to the generator, respectively.

**Table 6.** The influence of parameters in bicubic structured interpolation on classification accuracy of Cora.

| Parameter | Cora | Parameter | Citeseer |
|-----------|------|-----------|----------|
| g = 1369 | 84.1 | g = 3600 | 72.9 |
| g = 1444 | 84.5 | g = 3721 | 73.3 |
| g = 1521 | 78.6 | g = 3844 | 72.8 |

In Figure 6, we obtain the classification accuracy on Planetoid citation networks and the Cora, Citeseer, and Pubmed datasets. Specifically, as shown in Table 7, we compare the classification accuracy of Cora, Citeseer, and Pubmed datasets in different samples. The number of samples *s* is 4208, 4708, 5208, 5708, 5906, 6200, 6800, 7200,7800, 8200, 8800, 21,200, 22,500, and 23,200, respectively, and the corresponding classification accuracy on the dataset Cora is 78.6%, 82.3%, 84.5%, 84.1%, 83.8%, 82.8%, 81.6%, 81.1%, 79.8%, 78.8%, 76.5%, 75.8%, 76.8%, 75.2%; the corresponding classification accuracy on the dataset Citeseer is 70.6%, 72.1%, 72.6%, 72.9%, 71.3%, 70.2%, 68.8%, 67.9%, 73.3%, 64.5%, 62.8%, 60.9%, 59.8%, 57.8%; and the corresponding classification accuracy on the dataset Pubmed is 80.9%, 81.7%, 80.7%, 81.8%, 80.8%, 82.3%, 78.8%, 76.5%, 74.8%, 71.5%, 69.5%, 68.7%, 67.5%, 66.9%, respectively.

**Table 7.** Compare the performance of different sample sizes on different datasets Cora, Citeseer, and Pubmed.

| Parameter | Cora | Citeseer | Pubmed |
|-----------|------|----------|--------|
| s = 4208 | 78.6 | 70.6 | 80.9 |
| s = 4708 | 82.3 | 72.1 | 81.7 |
| s = 5208 | 84.5 | 72.6 | 80.7 |
| s = 5708 | 84.1 | 72.9 | 81.8 |
| s = 5906 | 83.8 | 71.3 | 80.8 |
| s = 6200 | 82.8 | 70.2 | 82.3 |
| s = 6800 | 81.6 | 68.8 | 78.8 |
| s = 7200 | 81.1 | 67.9 | 76.5 |
| s = 7800 | 79.8 | 73.3 | 74.8 |
| s = 8200 | 78.8 | 64.5 | 71.5 |
| s = 8800 | 76.5 | 62.8 | 69.5 |
| s = 21,200 | 75.8 | 60.9 | 68.7 |
| s = 22,500 | 76.8 | 59.8 | 67.5 |
| s = 23,200 | 75.2 | 57.8 | 66.9 |

We find that the Cora dataset has the highest classification accuracy when $s = 5208$, the Citeser dataset has the highest classification accuracy when $s = 7800$, and the Pubmed dataset has the highest classification accuracy when $s = 6200$. With the increase of the number of samples, the classification accuracy of the three datasets increases, but when the number of samples reaches a certain value, the accuracy value begins to decrease.

### 4.4.4. Robustness Analysis

We investigate the robustness of DGCGAN by constructing a disturbance graph against attack. Random attacks disturb the graphic structure by randomly adding false edges. Figure 7b,d,f show the classification accuracy of the random attack method for perturbation rate on the Cora, Citeseer, and Pubmed datasets, respectively. We have observed that DGCGAN consistently outperforms GCN, GAT, and IDGL. As shown in Figure 7b, when adding 15% stochastic edges to Cora, we discover that the classification accuracy of DGCGAN only decreased by 8.7%, while it decreased 11.9% for GCN, 39.9% for GAT, and 31.9% for IDGL.

In addition, as shown in Figure 7a,c,e, we study the robustness of DGCGAN by generating a disturbance graph against the attack on datasets Cora, Citeseer, and Pubmed. Metattack attacks the graph via deleting or adding edges on the basis of meta learning.

Under Metattack, the gap between DGCGAN and GCN/GAT/IDGL also increases with the perturbation rate.

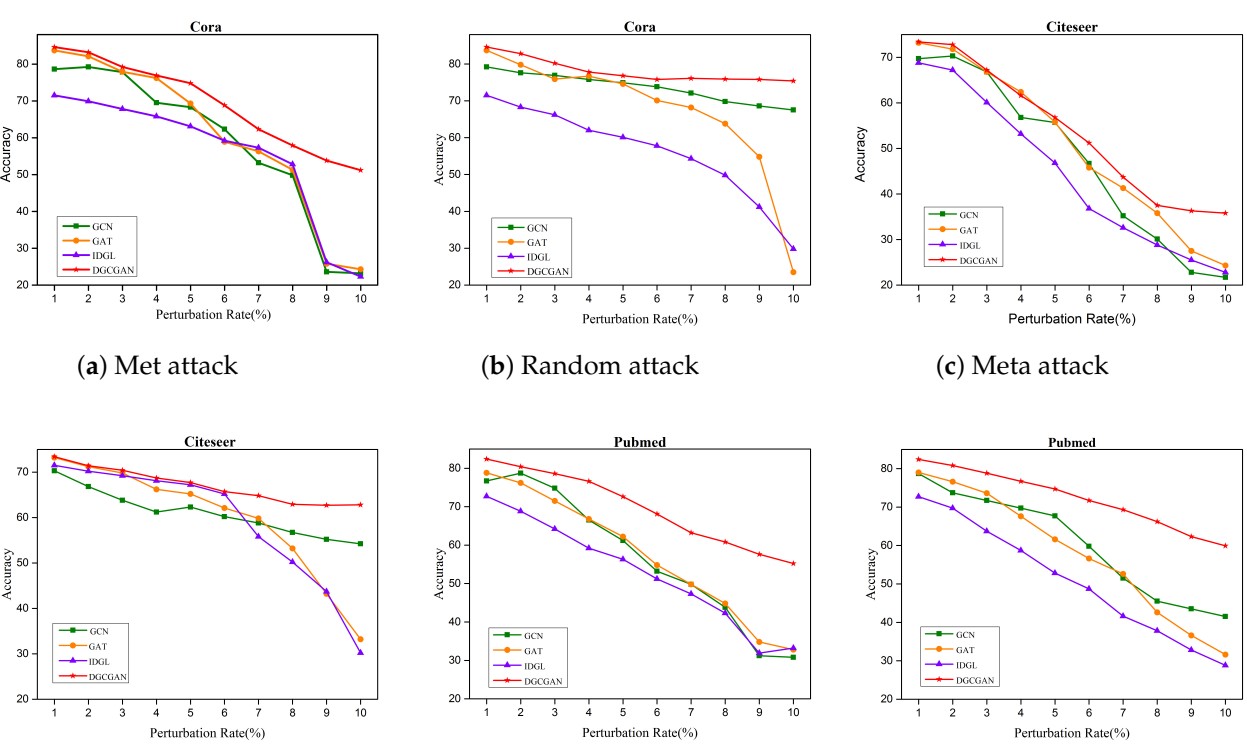

**Figure 7.** The robustness analysis on different datasets Cora, Citeseer, and Pubmed, in which (**a**,**b**) are robustness analysis on tbe Cora dataset, (**c**,**d**) are robustness analysis on the Citeseer dataset, and (**e**,**f**) are robustness analysis on the Pubmed dataset.

## 5. Conclusions

To solve the problem that the classification accuracy of GCN cannot be improved when the number of samples is small and the models lack robustness, In this paper, we develop a novel GCN model, the DGCGAN model, for semi-supervised classification of graphic structure data.The DGCGAN model is proposed to expand the number of samples and also enhances the robustness of the model due to the fusion of DCGAN. The node features will be obtained by symmetric normalized Laplace transformation on the feature matrix and adjacency matrix of the graph. In addition, the node features are extended to the regularly structured data (such as images) via the bicubic structured interpolation method. Then, after the feature-enhanced module, the new richer and more representative node features are obtained, which will be sent to the discriminator as positive samples along with the negative samples produced by the generator. Finally, these positive and negative samples are classified. The effectiveness of the proposed DGCGAN mode is demonstrated in several state-of-the-art methods for the semi-supervised node-classification task. Especially when the labeled data is extremely scarce, the proposed method can also obtain more significant results.

**Author Contributions:** Conceptualization, N.J.; methodology, N.J.; software, W.G.; investigation, L.J.; resources, X.T.; data curation, N.J.; writing—original draft, N.J.; writing—review and editing, N.J. All authors have read and agreed to the published version of the manuscript.

**Funding:** This work was supported by the National Natural Science Foundation of China under Grant 61977052.

**Data Availability Statement:** Not applicable.

**Conflicts of Interest:** The authors declare no conflict of interest.

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
