# Peer review of "Deep Graph-Convolutional Generative Adversarial Network for Semi-Supervised Learning on Graphs"

_remotesensing, doi:10.3390/rs15123172_

Round 1
Reviewer 1 Report
The paper developed a new model combining graph convolutional network and convolutional generative adversarial network to learn attribute and structure information on graph data. They used several open source datasets to test the proposed model and compared it with some basline models. Results show that it performed better than the compared basline models. However, there are also some issues.
1. There are many studies to integrate new deep learning techniques to achieve better results. But, its motivation for the integration should clarify. Why do you integrate the two models?
2. Table 2 and Table 3 show the results for compared models. However, what's the used index for assessing the performance. Classification and clustering are different. The used index should be different. However, I cannot evaluate the performance from the two tables.
3. They only tested the proposed model on open source datasets. It should use some real data related to the special issue (Deep Learning and Big Data Mining with Remote Sensing) to verify its effectiveness.
Reviewer 2 Report
I have the following concerns.
1. In the tasks of machine learning, when comparing different classification methods, it is necessary to compare not only the accuracy of the classification, but also the time spent. This especially applies to remote sensing. This is missing in the article.
2. Is it sufficient to use a polynomial of the third degree for interpolation? Justify this.
3. How is the choice of symmetric normalized Laplace transform justified.
4. How does the classification accuracy of your method depend on the size of the training sample. Show it.
Minor editing of English language required.
Round 2
Reviewer 1 Report
Authors addressed all my concerns and it's ready for publication.
Reviewer 2 Report
I am satisfied with the answers to my concerns. The changes and additions made have significantly improved the perception of the results obtained in the article.
Minor editing of English language required